# Effect of Stroking on Serotonin, Noradrenaline, and Cortisol Levels in the Blood of Right- and Left-Pawed Dogs

**DOI:** 10.3390/ani11020331

**Published:** 2021-01-28

**Authors:** Mirosław Karpiński, Katarzyna Ognik, Aleksandra Garbiec, Piotr Czyżowski, Magdalena Krauze

**Affiliations:** 1Department of Animal Ethology and Wildlife Management, Faculty of Animal Sciences and Bioeconomy, University of Life Sciences in Lublin, Akademicka st. 13, 20-950 Lublin, Poland; miroslaw.karpinski@up.lublin.pl (M.K.); aleksandra.garbiec@up.lublin.pl (A.G.); piotr.czyzowski@up.lublin.pl (P.C.); 2Department of Biochemistry and Toxicology, Faculty of Animal Sciences and Bioeconomy, University of Life Sciences in Lublin, Akademicka st. 13, 20-950 Lublin, Poland; katarzyna.ognik@up.lublin.pl

**Keywords:** stroking stimuli, behavioral tests, neuropeptides, laterality, dogs

## Abstract

**Simple Summary:**

The endocrine balance, reflected in the level of neuromodulators, is necessary for maintenance of homeostasis and facilitates adaptation to stressful conditions. An important role in processing the information reaching the brain is played by the asymmetric specialization of cerebral hemispheres referred to as laterality. The domestic dog shows preference of the forelimb determined by the predominance of the activity of the right or left brain hemisphere. Investigations of various animal species indicate that the left-brain hemisphere is involved in the control of unchanging stimuli or repetitive actions, while the right hemisphere is specialized in the quality of emotional reactions such as aggression or fear. A skillful observation of combined behavioral and physiological symptoms of stress in dogs provides better insight in the dog’s perception of veterinary care, and any means of reduction of the stress level is highly recommended, as it improves animal’s welfare. The results of the present study indicate that dogs’ laterality and sex affect the stress response and stroking can relieve stress. The level of the analyzed neuroregulators indicating intensification of stress or adaptation to stress conditions was higher in the males and in the right-pawed dogs. Our results confirm our assumptions that right-pawed dogs are better adapted to stressful conditions.

**Abstract:**

It has been assumed that stroking relieves stress responses in dogs, and dogs with the activation of the left-brain hemisphere (right-pawed) may show better adaptation to stress conditions. The aim of the study was to determine whether the stroking stimulus induced changes in the level of selected neuroregulators in dogs’ blood and whether these changes depended on the sex and the predominance of the activity of one of the brain hemispheres. The study involved 40 dogs of various breeds and both sexes. The experimental animals were subjected to a behavioral tests (Kong test), and the levels of noradrenaline, serotonin, and cortisol were determined in their blood plasma. The results of the behavioral test revealed that most dogs exhibited increased activity of the left hemisphere. Furthermore, irrespective of the sex and paw preference, stroking the animal was found to alleviate the stress response, which was reflected in reduced cortisol levels and increased serotonin levels. It was found that the plasma noradrenaline, cortisol, and serotonin levels were lower in the female dogs than in the males. Additionally, the plasma noradrenaline and serotonin levels were higher in the right-pawed dogs than in the left-pawed dogs. The present results confirm the assumption that right-pawed dogs adapt to stressful conditions more readily.

## 1. Introduction

A dog’s behavior is a function of genetic, epigenetic, and environmental factors [1,2]. The endocrine balance reflected in the level of neuromodulators such as dopamine and serotonin as well as cortisol or noradrenaline is necessary for maintaining homeostasis and facilitates adaptation to stressful conditions [3,4]. Problems with dog’s hyperactivity, aggression, or excessive fearfulness are a result of the additive effect of various factors, e.g., the presence of another individual and socialization errors [5,6]. An important role in processing the information reaching the brain is played by the asymmetric specialization of cerebral hemispheres referred to as laterality. The asymmetric specialization of the cerebral hemispheres known as lateralization/sidedness plays an important role in the processing of information reaching the brain. Brain asymmetries are possibly an evolutionary adaptation improving the brain function by allowing it to perform more than one task at a time. These asymmetries are referred to as laterality, which may vary depending on the complexity of tasks and/or organs. In recent years, a great body of evidence has been collected on not only structural but also functional and behavioral laterality in humans, many species of other vertebrates, and invertebrates. Analysis of brain laterality may be useful as part of a cognitive approach to the study of animal emotional processing. Studying motor laterality and understanding animals’ emotions is essential for improvement of animal welfare. In animals, emotional states are usually recognized with the use of behavioral and physiological measurements [7,8]. Laterality is a frequently reported phenomenon studied both in humans and in various animal species [9,10,11,12].

Human–dog interactions involve various types of sensory stimulation, e.g., tactile, auditory, visual, and olfactory stimuli. It has been shown that tactile caressing stimulation combined with talking to the dog can contribute to reduction in the level of cortisol in the organism. Furthermore, affiliate interactions such as stroking, talking, playing, and obedience training have been found to reduce physiological and behavioral stress responses in e.g., dogs in shelters [13,14].

The dogs’ sense of security is primarily influenced by the relationship with their owners, which should be based on trust and sense of comfort in their presence [15]. The owner-dog relationship is believed to exhibit behavioral and neuroendocrine similarities to the relationship between mothers and infants [16,17,18]. Any deviation from the daily routine, e.g., a visit to a veterinary clinic, is associated with an emotional (stress-inducing) response in the dog’s organism. In biological terms, emotions can be defined as a state of biochemical balance of the organism activating a specific “program” of response to external stimuli. A characteristic trait of emotions is that they are evoked suddenly and trigger somatic response. The neurotransmitter homeostasis in the organism is influenced by the ability to display natural behaviors resulting from the dog’s ethogram [19]. An appropriate balance between activity and rest is a determinant of homeostasis. Such a balance is extremely important, as stimulation triggers the release of excitatory neurotransmitters (dopamine, noradrenaline, glutamate) and activates the sympathetic nervous system. In turn, restful activities promote the release of gamma-aminobutyric acid and serotonin and stimulate the parasympathetic system. A skillful observation of combined behavioral and physiological symptoms of stress in dogs provides better insight in the dog’s perception of veterinary care, and any means of reduction of the stress level is highly recommended, as it improves animal’s welfare [20,21].

The domestic dog (*Canis familiaris*) playing or reaching for food most commonly shows preference of the forelimb determined by the predominance of the activity of the right or left brain hemisphere [22,23,24]. As indicated by Alves et al. [25], Shobe [26], and Harmon-Jones et al. [27], the activation of the left part of the brain is associated with stimuli that are pleasant to the dog, e.g., the owner approaching the animal, whereas the right part is activated by stressful stimuli, e.g., the presence of other dogs. Furthermore, as reported by Siniscalchi et al. [28], dogs wag their tails to the right when they see something they would like to approach and to the left when they see something they would prefer to avoid. Investigations conducted by Ogi et al. [21] demonstrated a positive effect of stroking on the emotional state of dogs. The results of research conducted by Field et al. [29] and Okabe et al. [30] indicate that stroking contributes to a decrease in the level of cortisol and an increase in the level of serotonin in dogs’ blood. As suggested by these researchers, such a response indicates mitigation of stress and an activating effect of stroking on the animal organism [29,30,31] demonstrated in their study that that shelter dogs stroked for 15 min clearly calmed down and the level of cortisol in their blood declined. As shown by DeVries et. al. [32] and Mariti et al. [33], stroking reduces the stress response and increases relaxation in dogs through oxytocin release. In contrast, Ogi et al. [21] and Lewandowski [34] suggest that stroking may also increase the blood cortisol level, which can be explained by the synchronization of organism stress, in agreement with the results reported by Pirrone et al. [35] and Butler et al. [36]. As suggested by Major et al. [37], stroking stimulates the nervous system through various mechanoreceptors located in the skin. Stroking activates signal transduction pathways along the nerve fibers and release of serotonin [38].

As reported by Wells et al. [39] left-pawed animals exhibit stronger fear responses, are more likely to show aggression, and are less able to cope with stressful situations than right-pawed animals. It has been assumed that stroking relieves stress reactions in dogs and that individuals with the predominance of the left hemisphere activity (right-pawed dogs) may exhibit greater adaptation to stress conditions. The aim of the study was to determine whether the stroking stimulus induced changes in the level of neuroregulators in the blood of dogs and whether these changes depended on the sex and the predominance of one of the brain hemispheres.

## 2. Methods

### 2.1. Animals and Procedures

The experiment involved 40 dogs (20 females and 20 males) of various breeds and different sizes, which were patients of one of the animal clinics in Lublin (Poland). In the group of dogs, 4 were pure breed (Jack Russel Terrier, Chow Chow and 2 Labradors), the other were hybrids (without breeding documentation). As for the size of the body, the weight range ranged from 4.5 kg to 22 kg. All behavioral and veterinary activities were performed by a 2-person team consisting of a veterinarian and a veterinary behaviorist technician. The appointments were always made at 9 o’clock, i.e., before the regular clinic opening time to avoid the presence of other patients. The selected animals lived in urban households (blocks of flats, houses). The mean age of the dogs included in the experiment was 3.91 years, while the Standard error of the mean (SEM) value was 0.39. All owners consented to the inclusion of their dogs in the experiment. The animals were healthy and non-sterilized. They had been provided with appropriate anti-parasitic and anti-infective prophylaxis. None of the dogs included in the study had received previous behavioral training. The interview with the owner prior to the veterinary examination did not indicate any behavioral disorders, e.g., excessive aggression or fearfulness. The visit to the veterinary clinic was associated with routine medical treatments. The animals were to undergo subsequent sterilization or oral cavity hygiene procedures under pharmacological anesthesia. As part of the medical veterinary treatments, the dogs were subjected to routine clinical examinations, which included the analysis of hematological and biochemical blood markers.

Blood was collected from *vena cephalica*. Since the blood collection was part of the standard veterinary treatments, no consent from the local animal experimentation commission was required. Before the first blood sampling the dog was isolated from the owner (applied 5–10 min of isolation, it was a negative factor, NS = negative stimuli) in a kennel cage placed in a windowless 3 × 3 m room for 5 min. Then, in the veterinary office, it was performed the activities related to the assessment of the animal’s condition, clinical examination and preparation for blood collection. The next step was the blood sampling, and it was treated as the “zero” sample. The duration of these activities were next 5 min. The analysis of hematological and biochemical blood markers performed routinely before the start of the experiment showed that their values in the blood of all experimental dogs were in the range considered as a reference in dogs [40]. The general medical-veterinary diagnosis showed that the dogs were healthy; hence, they were qualified for the experiment. The dog’s second visit to the veterinary office (2–4 weeks later after first) began with asking each of owners were asked to stroke their dogs for 5–10 min on the neck and forechest area and speak gently to the animal (it was a positive stimuli = SS). These activities were performed in the waiting room. In this case, the stroking time was determined individually and ranged from 5 to 10 min depending on the behavior of the dog, i.e., such behavioral symptoms as licking, turning the head away, or restlessness (some dogs needed 2–3 min to calm down). The effective stroking and gentle speaking time was 5 min. Then, in the veterinary office (without the presence of the owner) blood was collected. Simultaneously, it was the second, and final blood sampling from a patient used for the purposes of this study. Immediately afterwards, it was joined the planned sterilization procedure or dental surgery.

### 2.2. Behavioral Tests—Paw Preference Tests

#### 2.2.1. Kong Test

The Kong behavioral test (Kong ball test) is used to determine motor laterality (paw preference). The Kong test is based on the use of a toy dedicated for this purpose, i.e., the so-called Kong™ ball (KONG Company, Golden, CO, USA), which is commonly used for assessment of motor asymmetry in domestic dogs. The toy is a hollow conical rubber ball moving in a characteristic irregular way [39,41,42,43,44]. A medium-size Kong ball (approx. 11 cm) was used in the test. The Kong ball has holes at each end: one with a diameter of approx. 3 cm and the other with a diameter of 1 cm. To encourage the dog to use the paw repeatedly, a treat, most often dog’s favorite wet food, was loaded through the larger hole. Before starting the test, the dog was allowed to sniff the ball (Kong), which was then placed directly in front of the assessed individual.

Before taking the Kong test, it was performed the observations were intended to assess the degree of socialization of the dogs and their response to the presence and touch of a stranger. To this end, a headband was routinely worn over the dog’s snout. A voice stimulus was used to enhance the effect of the tactile stimulus.

The next step was to start the activities related Kong test, to the motor laterality associated with brain hemisphere activity was assessed in the dogs. The study was divided into two parts. To prevent fatigue/routine in the tested dogs, the first part of the test was performed during the first visit at the veterinary surgery at the time of waiting for the laboratory test results. For this purpose, the maximum number of repetitions was performed, usually about 20–30.

In turn, the second part of the test was carried out in the dog’s home environment, between the first and second visits (done 100 repetitions). The owners were given instructions on how to perform the test and informed about the need to record a video film.

In each part of the test, the front paw used by the dog to stabilize the toy in an attempt to reach the treat was recorded. Only clearly lateralized uses of paws (left or right paw) were counted. Cases of the use of both paws simultaneously were not counted. Each attempt to hold the ball with either the right or left paw by each dog (in total 100 times) was recorded.

#### 2.2.2. Paw Preference Index (z)

The first 100 (Kong test) L or R paw scores were used to calculate a binomial *z score* for each dog to determine whether the paw preference differed significantly from chance. The formula used to calculate it was z = (R − 0.5N)/√(0.25N), where R signifies the number of R paw uses and N signifies the sum of L plus R paw uses. Dogs with a positive z score value equal to or greater than 1.96 were R-pawed, those with a negative z score value equal to or less than −1.96 were L-pawed, and the other dogs were ambilateral A (showing no paw preference). A handedness index (HI) was also calculated for each dog (L − R/L + R); hence, a score of 1.0 represents exclusive use of the L paw and −1.0 denotes exclusive use of the R paw. The absolute value of HI is the strength of paw preference with the highest possible value of 1.0 indicating the exclusive use of either the L or R paw. A HI score of 0 indicates equal use of the L and R paws [45].

### 2.3. Blood Analysis—Determination of the Neurotransmitters Level

The level of cortisol was determined in blood serum using a Canine cortisol Elisa Kit produced by Cusabio Biotech. Ltd. (Wuhan, China). The level of noradrenaline was determined in the blood serum using a NA/NE (noradrenaline/norepinephrine) Elisa Fine Test (Wuhan, China). The level of serotonine was determined using ST/5-HT (5-hydroxytryptamine) Fine Test (Wuhan, China) in the same material.

### 2.4. Statistical Analysis

This experiment was performed in a completely randomized 2 factorial design, and the data (presented as the mean ± standard error of the mean) were subjected to two-way ANOVA to examine the effect of the stroking stimulus on changes in the level of neuroregulators in the blood of dogs and to determine whether these changes depend on the sex and the predominance of activity of one of the brain hemispheres. The Shapiro–Wilk and Levene tests were applied to test the model assumptions of normality and homogeneity of variance. The significance level was set at *p* = 0.05, and statistical calculations were performed using the General Linear Model (GLM) procedures of the STATISTICA software system ver. 12.0 (StatSoft Inc., Tulsa, OK, USA, 2014).

## 3. Results

### 3.1. Paw Preference Index (z)

The calculations of the paw preference index (z) revealed that all the dogs had a clear preference for using exclusively one of the paws (no ambilateral animals), as evidenced by the range of the paw preference index values. Dogs with a positive z score were R-pawed, those with a negative z score were L-pawed. The results of our experiment showed that 8 individuals of each sex exhibited left paw preference and 12 individuals of each sex had right paw preference (Figure 1).

These results were also confirmed by the calculated strength of preference HI, whose mean absolute value exceeded 0.5 in both the right- and left-pawed individuals (Table 1).

### 3.2. Biochemical Blood Indices

Regardless of the sex and right- or left- pawed laterality, the blood plasma of the stroked animals exhibited elevated levels of noradrenaline (*p =* 0.011 and *p =* 0.049) and serotonin (*p =* 0.023 and *p =* 0.042) and a reduced cortisol level (*p =* 0.031 and *p =* 0.026) (Table 2 and Table 3). The two-factor analysis of variance showed lower levels of noradrenaline, serotonin, and cortisol in the blood plasma of females in comparison with males (*p =* 0.042, 0.032, 0.042) (Table 2). A higher concentration of noradrenaline (*p =* 0.042) and serotonin (*p =* 0.049) was found in the blood plasma of the right-pawed dogs than in the left-pawed individuals (Table 3).

## 4. Discussion

A visit to the veterinary surgery may trigger a stress response, which is reflected in behavioral reactions and hormonal changes in dog’s blood. Biological measurements of stress include the analysis of the level of stress hormones, which are proportional to the stress level [20,46,47]. As suggested by Cozzi et al. [48], the isolation of the dog from its owner during veterinary examination may be one of the causes of reinforcement of stress-related behaviors and increased blood cortisol concentrations [15]. The present study has shown that dogs’ stress responses can be alleviated by stroking, as indicated by the lower cortisol levels and elevated serotonin levels. The reference blood cortisol level in dogs is 27.6–165.5 nmol/L, and the serotonin concentration ranges from 387.4 to 510.2 ng/mL [38,49,50,51]. Serotonin inhibits impulsive animal behavior, regulates metabolism, and influences signal transduction between neurons [52], whereas cortisol should be associated with long-term stress related to disturbance of homeostasis [53]. As shown by Berman et al. [54], excessively low serotonin levels may lead to a generalized state of hyperactivity and reduction of the sensitivity threshold at which the animal responds to provocative stimuli. As reported by Çakiroglu et al. [55], the serum of aggressive dogs exhibits a lower level of serotonin than that in animals showing no signs of aggression. Diego et al. [56], Field et al. [57], and Major et al. [37] show that stroking reduces noradrenaline release into the bloodstream via stimulation of skin receptors activating the limbic system. In the present study, we observed that stroking did not decrease but even increased the levels of noradrenaline in the dogs’ blood. As reported by Hart et al. [53] and Major et al. [37], short-term stress associated with an increase in the noradrenaline blood level. Many authors have shown that the neuronal and immune systems cooperate to maintain homeostasis [58,59,60,61]. 

As suggested by Bangasser et al. [62], the animal’s sex determines differences in the levels of stress hormones, and stress itself may indirectly influence the release of sex hormones. Our investigations have shown lower levels of noradrenaline and cortisol in females than in males; hence, the need of being stroked to calm the stress response is lower in females than in males. Additionally, we found higher blood serotonin levels in the males than in the females. In turn, McEwen et al. [63] suggest that females are more resistant to stress due to the presence of estrogen and the lower levels of cortisol and noradrenaline in their organisms. Major et al. [37] have observed that stroking contributes to a substantial reduction in noradrenaline blood levels in male dogs. As demonstrated by Hart et al. [53], blood noradrenaline levels in dogs are largely determined by their sex. The elevated noradrenaline levels in the blood of male dogs may be associated with the stimulating effect of testosterone on the activity of this enzyme [64]. In turn, estrogens activate catechol-o-methyltransferase (COMT), which is responsible for degradation of noradrenaline in the liver [53], thereby contributing to the reduction of the level of this neuropeptide in the blood of females.

The present study has shown that most of the analyzed dogs exhibit left hemisphere specialization and are therefore right-pawed. It is believed that Tan [65] was the first to publish the results of paw preference in dogs. As shown in his research, 57.1% of the studied population was right-preferent, 17.9% were left-preferent, and 25.0% were ambidextrous. The author does not report on differences between the sexes. In turn, Quaranta et al. [23] have reported that male dogs prefer the left front paw, whereas female dogs (*n* = 29) show preference for the right paw. In the study conducted by Wells et al. [45] ambilateral individuals predominated in the tape and Congo tests. As in the present study, there were more individuals with right paw preference than left-pawed animals, but the differences were not statistically significant.

As demonstrated by MacNeilage et al. [10], the right brain hemisphere is responsible for responses to strong emotions and stimulates an increase in cortisol release in the blood by controlling endocrine functions. In turn, the left hemisphere of the brain allows focusing on specific stimuli without distraction by signals that are not related to the performed task [10]. Literature data indicate a possible relationship between motor laterality and the emotional state of dogs, although it is believed that dogs with left hemisphere predominance are usually calmer than left-pawed individuals [9,66,67]. The noradrenaline and serotonin levels in the blood plasma of the right-pawed dogs were higher than in the left-pawed individuals. Based on these results, it can be assumed that right-pawed dogs are less susceptible to stress and adapt to stressful conditions more easily than left-pawed dogs. Studies conducted by such experts on canine brain asymmetry as Siniscalchi [68] and Quaranta [23] have demonstrated a relationship between paw preference and sex at the population level, as males exhibited a left paw preference and females preferred using the right paw. Similar results were presented by Wells [24] and McGreevy [22]. It should be emphasized that there are many studies in which no relationship between the sex of dogs and brain laterality was found [68,69,70]. There are also sex-related differences in the anatomy of the hippocampus, i.e., one of the most important centers of neurohormonal regulation. Post mortem studies have revealed a sex-related morphological asymmetry in the canine hippocampus (larger in males than in females) and paw preference. Left-pawed female dogs indeed had larger hippocampi than the right-pawed ones [71]. The issue of the influence of positive stimuli in the aspect of enhanced activity of one of the brain hemispheres and the level of neurohormones in dogs is still insufficiently investigated.

## 5. Conclusions

Our research suggests that by stroking your dog’s stress response can be reduced. Nevertheless, due to the significantly higher levels of serotonin and noradrenaline in the blood of right-pawed dogs, it can be assumed that these animals cope with stress better than left-pawed dogs. In order to finally confirm these assumptions, further research is needed on this subject.

## Figures and Tables

**Figure 1 animals-11-00331-f001:**
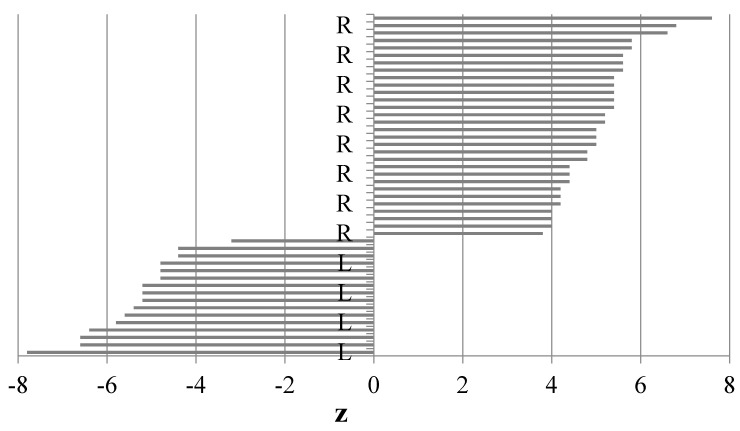
Ranges of the preference index (z) for both paws (based on the Kong test), L-left paw, R-right paw.

**Table 1 animals-11-00331-t001:** Range of the preference index (z) and absolute value of the strength of preference handedness index (HI) for both paws.

Item	z		HI (Absolute Value)	
	Mean	Range	Mean	Range
Left paw *n* = 16	−5.4	−7.8–3.2	0.54	0.3–0.8
Right paw *n* = 24	5.1	3.8–7.6	0.51	0.4–0.8

n—number of dogs.

**Table 2 animals-11-00331-t002:** Noradrenaline, serotonin, and cortisol levels in the blood serum of dogs in relation to the sex and the use of the stroking stimulus (*n* = 40).

Items		Noradrenaline	Serotonin	Cortisol
		pg/mL	ng/mL	ng/mL
F + NS		238.48	457.73	51.83
M + NS		278.52	602.51	59.16
F + SS		344.18	507.08	46.16
M + SS		451.33	578.41	48.97
SEM		0.039	0.073	0.012
Effect of sex (EP)	F	291.33 ^b^	482.41 ^b^	48.99 ^b^
	M	364.33 ^a^	590.46 ^a^	54.07 ^a^
Effect of stroking (EB)	NS	258.50 ^y^	530.12 ^y^	55.49 ^x^
	SS	397.76 ^x^	542.75 ^x^	47.57 ^y^
*p*-value				
EP effect		0.042	0.032	0.042
EB effect		0.011	0.023	0.031
EP x EB interaction		0.235	0.218	0.089

^x,y; a,b^ Means within the same column differ significantly (*p* ≤ 0.05) according to Tukey’s test (effect of sex (EP) x effect of stroking (EB) interaction); Standard error of the mean (SEM) divided by the square root of the replication number, *n* = 40; NS—negative stimuli, SS—positive stimuli; groups: F+NS—female dogs before stroking; M+NS—male dogs before stroking; F+SS—female dogs after stroking, M+SS—male dogs after stroking.

**Table 3 animals-11-00331-t003:** Noradrenaline, serotonin, and cortisol levels in the blood serum of dogs in relation to laterality and stroking effects (*n* = 40).

Item		Noradrenaline	Serotonin	Cortisol
		pg/mL	ng/mL	ng/mL
L + NS		147.38	501.95	59.65
R + NS		369.61	568.28	51.34
L + SS		367.25	527.65	43.5
R + SS		428.25	607.85	51.63
SEM		0.029	0.037	0.009
Effect of laterality (EL)	L	257.32 ^b^	514.80 ^b^	51.58
	R	398.93 ^a^	573.07 ^a^	51.49
Effect of stroking (EB)	NS	258.50 ^y^	541.75 ^y^	55.50 ^x^
	SS	397.75 ^x^	575.77 ^x^	47.57 ^y^
*p*-value				
EL effect		0.042	0.033	0.533
EB effect		0.049	0.042	0.026
EL x EB interaction		0.232	0.156	0.838

^x,y; a,b^ Means within the same column differ significantly (*p ≤* 0.05) according to Tukey’s test (effect of laterality (EL) x EB interaction). L—left, R—right, NS—negative stimuli, SS—stroking stimuli, SEM = SD divided by the square root of the replication number, *n* = 40. Groups: R+NS—left-pawed dogs before stroking; R+NS—right-pawed dogs before stroking; L+SS—left-pawed dogs after stroking, R+SS—right-pawed dogs after stroking.

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
