# Peer review of "Effect of Stroking on Serotonin, Noradrenaline, and Cortisol Levels in the Blood of Right- and Left-Pawed Dogs"

_animals, 2021, doi:10.3390/ani11020331_

Round 1
Reviewer 1 Report
This paper presents the results of studies examining what factors influence the levels of three hormones: noradrenaline, serotonin and cortisol in a stressful situation (blood sampling) in domestic dog. The level of the tested neuroregulators has been studied for gender, laterality, and the impact of dog stroking by owner.
The problem is very interesting and obtained results could significantly improve the welfare of dogs and reduce stress during treatments in veterinary clinics.
However, I have several serious objections related to the methodology of the experiment.
- The age range of the dogs participating in the experiment is very large (from 1 to 13 years), such variation may affect the result. The group of dogs should be less diverse and include individuals e.g. between 2 and 4 years of age.
- The authors declared that: “The interview with the owner prior to the veterinary examination did not indicate any behavioural disorders, e.g. excessive aggression or fearfulness”. However, was it checked before the experiment that all dogs responded positively to the touch / stroking of the owner? Not all dogs enjoy being touched to the same extent. In case of some individuals stroking can cause stress.
- The period of isolation/stroking the subjects has not been standardized (5-10 minutes). 10 minutes is 100% more than 5 minutes and such a difference can affect hormone levels.
- A third blood test (control group) should be performed. It should be carried out as soon as the dog arrives at the veterinary clinic (not after isolation or stroking)
Some additional questions and specific comments:
- Was blood sampling always done by the same person? The personality and behaviour of the person doing blood sampling can significantly affect the level of stress in animal.
- Whether the experiment was always conducted at the same time of day? For example, cortisol levels fluctuate throughout the day.
- There is no information in the manuscript about the number of males and females within right-pawed and left-pawed dogs separately. If there was a predominance of one sex in right-pawed or left-pawed dogs, this could affect the result.
- How was the dog isolated prior to blood sampling? Was the dog in the company of strangers or alone?
Line 33-34 “… reduced cortisol and noradrenaline levels…” – According to Table 2 and 3 stroking caused increasing noradrenaline level
Line 45-47 – Publication [4] is omitted
Line 53 – “Another highly important element is the ability to avoid distressing situations”. What did the authors mean?
Line 72 – Instead of [24] should be [23]
Line 89 – “None of the dogs included in the study had received previous behavioral training.” What kind of training did the authors mean?
Line 108 – “(stroking stimuli; PS = positive stimuli)”. I suppose that it should be SS instead PS (as in the tables)
Line 110-117 – It is better to move this paragraph to section 2.2.1
Line 136 – It should be 2.2.2 instead of 2.2.3.
Line 166-167 – “The preference index value was lower than -1.96 in all dogs assessed as left-166 pawed and greater than +1.96 in the right-pawed animals”. This information has already been given before (line 140-141).
Instead of it number of right-pawed and left-pawed females and males should be provided.
Line 189-190 – What does mean “females/males before stroking”? Was there another blood sampling were performed just before stroking, i.e. there were three examination in total?
Line 281 - According to Table 2 and 3 noradrenaline levels were increased.
Table 2 and 3
Results of “Effect of stroking (EB)” were given twice (in Table 2 and 3). I do not understand why the blood test results are partly the same and partly different.
I find the experiment very interesting and the results could be very valuable. However, due to significant methodological errors I don’t recommend this manuscript for publication in this form.
Author Response
COMMENTS TO THE REVIEWER
Thank you for the review; we have responded to the comments in the text. Fragments that have been modified or added are marked with a different color. Irrelevant references have been removed or replaced with appropriate ones. Missing references have been added.
Most of the revisions addressing the review have been made directly next to the Reviewer's comments in the text of the manuscript. They are marked in red.
Reviewer 1
Comments and Suggestions for Authors
This paper presents the results of studies examining what factors influence the levels of three hormones: noradrenaline, serotonin and cortisol in a stressful situation (blood sampling) in domestic dog. The level of the tested neuroregulators has been studied for gender, laterality, and the impact of dog stroking by owner.
The problem is very interesting and obtained results could significantly improve the welfare of dogs and reduce stress during treatments in veterinary clinics.
However, I have several serious objections related to the methodology of the experiment.
- Reviewer 1
The age range of the dogs participating in the experiment is very large (from 1 to 13 years), such variation may affect the result. The group of dogs should be less diverse and include individuals e.g. between 2 and 4 years of age.
Answer: In fact, it would be more accurate to leave only the mean age of 3.9 and the SEM of 0.39. Such an unfavorable effect of age discrepancy is ascribed to three dogs, one - 1.2 years old and two older dogs - 11 and 13 years old. The authors did not decide to remove these dogs from the experimental group, as the age range of the dogs included in the experiment was not regarded as a factor influencing the level of hormones and neurotransmitters, likewise the maintenance of the dog (block of flats, house). It was also assumed that age did not have an effect on paw preference. In the text we have included information that the mean age of the dogs included in the experiment was 3.91 years, while the SEM value was 0.39.
- The authors declared that: “The interview with the owner prior to the veterinary examination did not indicate any behavioural disorders, e.g. excessive aggression or fearfulness”. However, was it checked before the experiment that all dogs responded positively to the touch / stroking of the owner? Not all dogs enjoy being touched to the same extent. In case of some individuals stroking can cause stress.
Answer: The assumption made by the authors was to use the simplest and feasible stimuli that each owner could apply. Following observations of everyday life, stroking and mild vocalization were selected.
During the first clinical trial, the observations were intended to assess the degree of socialization of the dogs and their response to the presence and touch of a stranger. To this end, a headband was routinely worn over the dog's snout. A voice stimulus was used to enhance the effect of the tactile stimulus. Next, the Kong test was performed in a separate room at the veterinary clinic.
- The period of isolation/stroking the subjects has not been standardized (5-10 minutes). 10 minutes is 100% more than 5 minutes and such a difference can affect hormone levels.
Answer: The authors conducted the observations and tests as part of daily work of the veterinary clinic. The text of the manuscript has been supplemented to contain information that the dog was isolated from the owner in a kennel cage placed in a windowless 3x3 m room for 5 minutes. The assessment of the animal's condition, clinical examination, preparation for blood collection, and the collection took another 5 minutes. In total, the duration of the isolation from the owner was up to 10 minutes. It is likely that in strict laboratory conditions, this period could be standardized to an average of e.g. 6.5 minutes.
In turn, the stroking time was determined individually and ranged from 5 to 10 minutes depending on the behavior of the dog, i.e. such behavioral symptoms as licking, turning the head away, or restlessness (some dogs needed 2-3 minutes to calm down). The effective stroking and gentle speaking time was 5 minutes.
- A third blood test (control group) should be performed. It should be carried out as soon as the dog arrives at the veterinary clinic (not after isolation or stroking)
Answer: There has been some misunderstanding about the “zero” sample. In fact, there is no clear indication in the manuscript that the first blood sampling was performed immediately on arrival at the clinic and was treated as the “zero” sample vs. the stimulus (stroking) test sample. Thus, there was no need for an additional/redundant third blood sampling.
Reviewer 1 Some additional questions and specific comments:
- Was blood sampling always done by the same person? The personality and behaviour of the person doing blood sampling can significantly affect the level of stress in animal.
Answer: All behavioral and veterinary activities were performed by a 2-person team consisting of a veterinarian and a veterinary behaviorist technician.
- Whether the experiment was always conducted at the same time of day? For example, cortisol levels fluctuate throughout the day.
Answer: Yes - the appointments were always made at 9 o'clock, i.e. before the regular clinic opening time to avoid the presence of other patients.
- There is no information in the manuscript about the number of males and females within right-pawed and left-pawed dogs separately. If there was a predominance of one sex in right-pawed or left-pawed dogs, this could affect the result.
Answer: Certainly, these data will be added. The results of our experiment showed that 8 individuals of each sex exhibited left paw preference and 12 individuals of each sex had right paw preference. We have included this information in the Results section and in Table 1.
- How was the dog isolated prior to blood sampling? Was the dog in the company of strangers or alone?
Answer: the dog was isolated from the owner in a kennel cage placed in a windowless 3x3 m room for 5 minutes.
Reviewer 1 Line 33-34 “… reduced cortisol and noradrenaline levels…” – According to Table 2 and 3 stroking caused increasing noradrenaline level
Answer Line 33-34: The reviewer is right, and it was indeed a mistake, as the level of noradrenaline increased after the use of stroking in the experiment. The error has been corrected in the manuscript.
Reviewer 1 Line 45-47 – Publication [4] is omitted
Answer Line 45-47: Indeed, the Reviewer is right, the literature number was mixed up and the number 5 was used instead of 4. The error has been corrected in the text.
Reviewer 1 Line 53 – “Another highly important element is the ability to avoid distressing situations”. What did the authors mean?
Answer Line 53: This sentence is incomprehensible and is a mental shortcut. The sentence has been deleted from the text.
Reviewer 1 Line 72 – Instead of [24] should be [23]
Answer Line 72: Indeed, the reviewer is right: the technical error in the citation of the literature has been corrected in the text. Due to the addition of several new literature items and changes resulting from the redrafting of the text proposed by other Reviewers, the current numbering has changed.
Line 89 – “None of the dogs included in the study had received previous behavioral training.” What kind of training did the authors mean?
Answer: The statement mentioned by the reviewer “No previous behavioral training” means that the owner had not used help/services of a professional behaviorist in terms of training and behavioral therapy. The human-animal interaction developed naturally during the everyday life in the household.
Reviewer 1 Line 108 – “(stroking stimuli; PS = positive stimuli)”. I suppose that it should be SS instead PS (as in the tables)
Answer Line 108: The abbreviation used may be confusing. The authors meant the stroking stimulus. As recommended by the Reviewer, the abbreviation SS (as in the Tables) has been used in the text.
Reviewer 1 Line 110-117 – It is better to move this paragraph to section 2.2.1
Answer Line 110-117: As suggested by the Reviewer, the fragment from lines 110-117 in the original version of the manuscript has been placed in section 3.2.1.
Reviewer 1 Line 136 – It should be 2.2.2 instead of 2.2.3.
Answer Line 136: Yes, the reviewer is right: this should be 2. 2.2. instead of 2.2.3. The error has been corrected in the text.
Reviewer 1 Line 166-167 – “The preference index value was lower than -1.96 in all dogs assessed as left-166 pawed and greater than +1.96 in the right-pawed animals”. This information has already been given before (line 140-141).
Answer Line 166-167: The information on the “z” indicator has been corrected and information on the number of right and left-footed dogs participating in the experiment has been added.
Reviewer 1 Instead of it number of right-pawed and left-pawed females and males should be provided.
Answer: The text has been expanded to contain information on the number of right-pawed and left-pawed females and males. There were 8 females and males with left paw preference and 12 individuals of each sex with right paw preference.
The following sentence has been included in the text: The results of our experiment showed that 8 individuals of each sex exhibited left paw preference and 12 individuals of each sex had right paw preference. We have included this information in the Results section and in Table 1.
Reviewer 1 Line 189-190 – What does mean “females/males before stroking”? Was there another blood sampling were performed just before stroking, i.e. there were three examination in total?
Answer Line 189-190: The expression “females/males before stroking” refers to dogs that had the blood sampled at the first veterinary visit during which no stroking stimulus was applied. During the second visit preceded by application of the stroking stimulus, the blood was sampled the second time. We regarded the first blood sampling (i.e. before applying the stroking stimulus) as the control for the result of the second blood sampling, i.e. after the stroking stimulus was applied to the dog. In the text below the table, the information has been changed to female dogs before/after stroking and male dogs before/after stroking to highlight the difference between the dogs of each sex before and after application of the stroking stimulus.
Reviewer 1 Line 281 - According to Table 2 and 3 noradrenaline levels were increased.
Table 2 and 3
Answer Line 281: Yes, the reviewer is right, the level of noradrenaline after stroking increased rather than decreased as written erroneously. We assumed that the increased noradrenaline levels may be related to agitation, short-term stress, or expectation of a reward.
Reviewer 1 Results of “Effect of stroking (EB)” were given twice (in Table 2 and 3). I do not understand why the blood test results are partly the same and partly different.
Answer: Indeed, the results in Tables 2 and 3 for noradrenaline and cortisol levels are nearly identical. The identical value for these indicators is a coincidence, but it is a correct result of mathematical calculations based on different partial means. We were also surprised by the convergence of the values, but we have checked the results once again and they are calculated correctly. We agree that their similarity might suggest a mistake.
Reviewer 1 I find the experiment very interesting and the results could be very valuable. However, due to significant methodological errors I don’t recommend this manuscript for publication in this form.
Answer: We have tried our best to respond to the reviews. We hope that the revisions introduced in accordance with the Reviewer's suggestion will improve the quality of the manuscript and will qualify our work as worth publishing in Animals.
Reviewer 2 Report
General comments
This is an interesting study on effective ways to assess an reduce acute stress in dogs using our knowledge of behavioral laterality and the link with emotional and physiological stress. The manuscript, however, needs some work before it can be accepted for publication.
Introduction: this section needs the most work. The information provided is not exhaustive, lacks flow, key citations are missing and the objectives and research questions of the study are not supported by previous research and knowledge. I have provided more detailed comments and suggestions below.
Methods: why the authors decided not to have a control group (i.e. non-stroked group) to control for time/repeated measure factor? This is a limitation of the study, it should be discussed.
Discussion: some discussion points are missing and more references are needed to support the authors statements and conclusions. Again I have provided some suggestions for improvement below.
Detailed comments
L 10 [add commas] The endocrine balance, reflected in the level of neuromodulators, is necessary for…
L15-16 This sentence is very simplistic, emotional laterality is much more complex. think about something on the lines of “the left part of the brain may be associated…OR has been shown to activate mainly..”
L19-21 specify these are findings from the present study
L25 this statement is inaccurate, what does “predominance of the left brain hemisphere” mean? maybe you mean activation of the brain?
L29 to a behavioral test
L 29-31 And when did the stroking come into play?
L43 of neuromodulators such as
L44 for maintaining homeostasis
L42-66 this introduction is very fragmented, it jumps from one statement/concept to another without clear transitions. E.g. L42 “Dog's behavior is a function of genetic, epigenetic, and environmental factors” then jumps to endocrine balance, then mentions behavioral problems and jumps to a brief notion of laterality, then moves to dog-owner bond, then moves on to emotional stress and back to endocrine homeostasis. Further, these concepts are key for the understanding of the rationale of the study and they are not nearly as exhaustive as they should to provide support to the methods and hypothesis of the study (see comments below)
L47-49 how is laterality linked to the previous statement? How is it an important role, what is the mechanism/neuroscience behind the link between laterality and emotional stress/emotional reactivity. Relevant papers that should be cited and used to expand on this concept that is key to support your research question are e.g. Leliveld, L. M., Langbein, J., & Puppe, B. (2013). The emergence of emotional lateralization: evidence in non-human vertebrates and implications for farm animals. Applied Animal Behaviour Science, 145(1-2), 1-14; Barnard, S., Matthews, L., Messori, S., Podaliri-Vulpiani, M., & Ferri, N. (2016). Laterality as an indicator of emotional stress in ewes and lambs during a separation test. Animal cognition, 19(1), 207-214;
L50-53 one of the main interventions in this study is stroking. This paragraph should expand more on the effect of dog-human interaction on dog stress, and coping strategies. There are already some papers cited that can be used to expand on this concept (e.g. Ref 17, 40, 41, 47). Other relevant papers are: Coppola, C.L., Grandin, T. and Enns, R.M., 2006. Human interaction and cortisol: can human contact reduce stress for shelter dogs?. Physiology & behavior, 87(3), pp.537-541; Csoltova, E., Martineau, M., Boissy, A., & Gilbert, C. (2017). Behavioral and physiological reactions in dogs to a veterinary examination: Owner-dog interactions improve canine well-being. Physiology & behavior, 177, 270-281)
L74-76 this needs expanded. What do you mean by adaptive strategy, how is that linked to laterality. Provide examples from literature.
L77-78 is there any evidence in the literature? See previous comment, but also caution the reader on possible negative effects of human contact as for some dogs (think not well socialized dog) this interaction could actually be aversive (see e.g. Kuhne, F., Hößler, J. C., & Struwe, R. (2014). Behavioral and cardiac responses by dogs to physical human–dog contact. Journal of Veterinary Behavior, 9(3), 93-97; Kuhne, F., Hößler, J. C., & Struwe, R. (2012). Effects of human–dog familiarity on dogs’ behavioural responses to petting. Applied animal behaviour science, 142(3-4), 176-1810
L85 add Country.
L111 Provide key literature used for this standard test
L110-117 this explanation is inconsistent: first it is state that motor laterality was assessed during the first visit, but then it is stated that there were 2 test sessions not one. Clarify if this was one test split in two sessions (i.e. leading to a cumulative final score) or two different tests (i.e. repeated measure, the dog was scored twice at different time points).
L137-146 provide reference of other studies using this same approach to support this statistical choice.
L209-237 some of this should be moved to the introduction (see previous comments)
L256-257 technically, this study demonstrated that most dogs in this population were right-pawed, hence you can assume they had a higher left-brin specialization..not the other way around.
L63-64 this sentence introduces the concept of temperament which has not been mentioned before. Either expand on why this may be relevant or delete.
L267-268 what is this assumption based on? Provide logical thought process and references.
L272 there is a body of dog literature that did not find this sex difference in lateral behavior, this should be acknowledged (e.g. Barnard, S., Wells, D. L., Hepper, P. G., & Milligan, A. D. S. (2017, April 17). Association Between Lateral Bias and Personality Traits in the Domestic Dog (Canis familiaris). Journal of Comparative Psychology; Branson, N. J., & Rogers, L. J. (2006). Relationship between paw prefer- ence strength and noise phobia in Canis familiaris. Journal of Compar- ative Psychology, 120, 176–183; Schneider, L. A., Delfabbro, P. H., & Burns, N. R. (2013). Temperament and lateralisation in the domestic dog (Canis familiaris). Journal of Veterinary Behavior: Clinical Applications and Research, 8, 124–134)
L255-257 a comparison with previous studied populations (i.e. did other study fing the same R vs L vs A distribution?) should be briefly discussed here.
Author Response
COMMENTS TO THE REVIEWER
Thank you for the review; we have responded to the comments in the text. Fragments that have been modified or added are marked with a different color. Irrelevant references have been removed or replaced with appropriate ones. Missing references have been added.
Most of the revisions addressing the review have been made directly next to the Reviewer's comments in the text of the manuscript. They are marked in blue.
Reviewer 2
Comments and Suggestions for Authors
General comments
This is an interesting study on effective ways to assess an reduce acute stress in dogs using our knowledge of behavioral laterality and the link with emotional and physiological stress. The manuscript, however, needs some work before it can be accepted for publication.
Introduction: this section needs the most work. The information provided is not exhaustive, lacks flow, key citations are missing and the objectives and research questions of the study are not supported by previous research and knowledge. I have provided more detailed comments and suggestions below.
Methods: why the authors decided not to have a control group (i.e. non-stroked group) to control for time/repeated measure factor? This is a limitation of the study, it should be discussed.
Discussion: some discussion points are missing and more references are needed to support the authors statements and conclusions. Again I have provided some suggestions for improvement below.
Detailed comments
Reviewer 2 L 10 [add commas] The endocrine balance, reflected in the level of neuromodulators, is necessary for…
Answer Line 10: This linguistic suggestion has been introduced in the text.
Reviewer 2 L15-16 This sentence is very simplistic, emotional laterality is much more complex. think about something on the lines of “the left part of the brain may be associated…OR has been shown to activate mainly..”
Answer Line 15-16: In accordance with the Reviewer's recommendation, the following sentence has been added: Investigations of various animal species indicate that the left brain hemisphere is involved in the control of unchanging stimuli or repetitive actions, while the right hemisphere is specialized in the quality of emotional reactions such as aggression or fear.
Reviewer 2 L19-21 specify these are findings from the present study
Answer Line 19-21: Information has been added that the results of the present study indicate that dogs' laterality and sex affect the stress response and stroking can relieve stress. The level of the analyzed neuroregulators indicating intensification of stress or adaptation to stress conditions was higher in the males and in the right-pawed dogs.
Reviewer 2 L25 this statement is inaccurate, what does “predominance of the left brain hemisphere” mean? maybe you mean activation of the brain?
Answer Line 25: As suggested by the Reviewer, the word “predominance” has been replaced by “activity”.
Reviewer 2 L29 to a behavioral test
Answer line 29: This linguistic suggestion has been introduced in the text.
Reviewer 2 L 29-31 And when did the stroking come into play?
Answer Line 29-30: The dogs were stroked during the second visit to the veterinary clinic immediately before the second blood sampling. Detailed information about the time of application of the stroking stimulus and the time sequence of this activity can be found in the Material and methods section.
Reviewer 2 L43 of neuromodulators such as
Answer Line 43: This linguistic suggestion has been introduced in the text.
Reviewer 2 L44 for maintaining homeostasis
Answer Line 44: This linguistic suggestion has been introduced in the text.
Reviewer 2 L42-66 this introduction is very fragmented, it jumps from one statement/concept to another without clear transitions. E.g. L42 “Dog's behavior is a function of genetic, epigenetic, and environmental factors” then jumps to endocrine balance, then mentions behavioral problems and jumps to a brief notion of laterality, then moves to dog-owner bond, then moves on to emotional stress and back to endocrine homeostasis. Further, these concepts are key for the understanding of the rationale of the study and they are not nearly as exhaustive as they should to provide support to the methods and hypothesis of the study (see comments below)
Answer: The apparent fragmentation may be related to the fact that we tried to combine the difficult subject of endocrinology with brain laterality. As for the hypothesis, we were inspired by the results of research presented by Wells et al. (2017), which suggest that left-pawed animals show stronger responses to fear, are more likely to show aggression, and are less able to cope with stressful situations than right-pawed animals.
Wells, D.L.; Hepper, P.G.; Milligan, A.D.S.; Barnard, S. Cognitive bias and paw preference in the domestic dog (Canis familiaris). Journal of Comparative Psychology, 2017, 131, 317-325. https://doi.org/10.1037/com0000080.
Reviewer 2 L47-49 how is laterality linked to the previous statement? How is it an important role, what is the mechanism/neuroscience behind the link between laterality and emotional stress/emotional reactivity. Relevant papers that should be cited and used to expand on this concept that is key to support your research question are e.g. Leliveld, L. M., Langbein, J., & Puppe, B. (2013). The emergence of emotional lateralization: evidence in non-human vertebrates and implications for farm animals. Applied Animal Behaviour Science, 145(1-2), 1-14; Barnard, S., Matthews, L., Messori, S., Podaliri-Vulpiani, M., & Ferri, N. (2016). Laterality as an indicator of emotional stress in ewes and lambs during a separation test. Animal cognition, 19(1), 207-214;
Answer Line 47-49: We appreciate for the suggestion of citing very relevant literature items. After a thorough analysis of the paper, we have added appropriate information in the text. The following fragment has been added in the manuscript: “The asymmetric specialization of the cerebral hemispheres known as lateralization/sidedness plays an important role in the processing of information reaching the brain. Brain asymmetries are possibly an evolutionary adaptation improving the brain function by allowing it to perform more than one task at a time. These asymmetries are referred to as laterality, which may vary depending on the complexity of tasks and/or organs. In recent years, a great body of evidence has been collected on not only structural but also functional and behavioral laterality in humans, many species of other vertebrates, and invertebrates. Analysis of brain laterality may be useful as part of a cognitive approach to the study of animal emotional processing. Studying motor laterality and understanding animals' emotions is essential for improvement of animal welfare. In animals, emotional states are usually recognized with the use of behavioral and physiological measurements (Leliveld, Langbein, and Puppe, 2013). As suggested by Barnard et al. (2016), laterality can be help to identify emotional processes and stress responses in animals.”
Leliveld, L.M.C.; Langbein, J.; Puppe, B. The emergence of emotional lateralization: Evidence in non-human vertebrates and implications for farm animals, Appl. Anim. Behav. Sci., 2013, 145, 1-14. doi.org/10.1016/j.applanim.2013.02.002.
Barnard, S.; Matthews, L.; Messori, S.; Podaliri-Vulpiani, M.; Ferri, N. Laterality as an indicator of emotional stress in ewes and lambs during a separation test. Anim. Cogn. 2016, 19, 207-214. doi: 10.1007/s10071-015-0928-3.
Reviewer 2 L50-53 one of the main interventions in this study is stroking. This paragraph should expand more on the effect of dog-human interaction on dog stress, and coping strategies. There are already some papers cited that can be used to expand on this concept (e.g. Ref 17, 40, 41, 47). Other relevant papers are: Coppola, C.L., Grandin, T. and Enns, R.M., 2006. Human interaction and cortisol: can human contact reduce stress for shelter dogs?. Physiology & behavior, 87(3), pp.537-541; Csoltova, E., Martineau, M., Boissy, A., & Gilbert, C. (2017). Behavioral and physiological reactions in dogs to a veterinary examination: Owner-dog interactions improve canine well-being. Physiology & behavior, 177, 270-281)
Answer Line 50-53: We thank the Reviewer for the suggestions, as they indicated relevant content that should be added in the manuscript. A new fragment has been added: “Human-dog interactions involve various types of sensory stimulation, e.g. tactile, auditory, visual, and olfactory stimuli. It has been shown that tactile caressing stimulation combined with talking to the dog can contribute to reduction in the level of cortisol in the organism. Furthermore, affiliate interactions such as stroking, talking, playing, and obedience training have been found to reduce physiological and behavioral stress responses in e.g. dogs in shelters (Coppola et al., 2006; Csoltova et al., 2017)”.
Coppola, C.L.; Grandin, T.; Enns, R.M. Human interaction and cortisol: can human contact reduce stress for shelter dogs?. Physiol. Behav. 2006, 87, 537-541. doi: 10.1016/j.physbeh.2005.12.001.
Csoltova, E.; Martineau, M.; Boissy, A.; Gilbert, C. Behavioral and physiological reactions in dogs to a veterinary examination: Owner-dog interactions improve canine well-being. Physiol. Behav. 2017, 177, 270-281. doi: 10.1016/j.physbeh.2017.05.013.
Reviewer 2 L74-76 this needs expanded. What do you mean by adaptive strategy, how is that linked to laterality. Provide examples from literature.
Answer Line 74-76: The sentence has been deleted, as it was found not to be relevant at this point. The information that was intended to be presented in the sentence cited earlier was focused on the possibility of dog adaptation to a stressful situation. The Reviewer's earlier suggestions helped us to develop the idea more precisely in the earlier part of the manuscript.
Reviewer 2 L77-78 is there any evidence in the literature? See previous comment, but also caution the reader on possible negative effects of human contact as for some dogs (think not well socialized dog) this interaction could actually be aversive (see e.g. Kuhne, F., Hößler, J. C., & Struwe, R. (2014). Behavioral and cardiac responses by dogs to physical human–dog contact. Journal of Veterinary Behavior, 9(3), 93-97; Kuhne, F., Hößler, J. C., & Struwe, R. (2012). Effects of human–dog familiarity on dogs’ behavioural responses to petting. Applied animal behaviour science, 142(3-4), 176-1810
Answer Line 77-78: Yes, similar indications have already been presented in the scientific literature and our hypothesis was based on the results of research conducted by Wells et al. (2017), which suggest that left-pawed animals show stronger fear responses, are more likely to show aggression, and are less able to cope with stressful situations than right-pawed animals. The latter, in turn, rely more on the left hemisphere of the brain in processing sensory information.
Wells, D.L.; Hepper, P.G.; Milligan, A.D.S.; Barnard, S. Cognitive bias and paw preference in the domestic dog (Canis familiaris). Journal of Comparative Psychology, 2017, 131, 317-325. https://doi.org/10.1037/com0000080.
Reviewer 2 L85 add Country.
Answer line 85: The information has been added: the study was performed in Poland.
Reviewer 2 L111 Provide key literature used for this standard test
Answer: Reference literature has been provided in subsection 2.2.1. Kong test.
Reviewer 2 L110-117 this explanation is inconsistent: first it is state that motor laterality was assessed during the first visit, but then it is stated that there were 2 test sessions not one. Clarify if this was one test split in two sessions (i.e. leading to a cumulative final score) or two different tests (i.e. repeated measure, the dog was scored twice at different time points).
Answer: The methodology has been completed with the following information: The test was performed in two sessions. The first session took place in the clinic during the first visit with the maximum number of repetitions (usually up to 20-30), and the second session was conducted as a continuation of the test in the owner's house to reach up to 100 repetitions.
Reviewer 2 L137-146 provide reference of other studies using this same approach to support this statistical choice.
Wells, D.L.; Hepper, P.G.; Milligan, A.D.S.; Barnard, S. Stability of motor bias in the domestic dog, Canis familiaris Behavioural Processes Volume 149, 2018, Pages 1-7.
Reviewer 2 L209-237 some of this should be moved to the introduction (see previous comments)
Answer: As suggested by the Reviewer, the Discussion and Introduction have been reedited.
Reviewer 2 L256-257 technically, this study demonstrated that most dogs in this population were right-pawed, hence you can assume they had a higher left-brin specialization..not the other way around.
Answer Line 256-257: Yes, the reviewer is right, most of the dogs in the study population were right-pawed.
Reviewer 2 L63-64 this sentence introduces the concept of temperament which has not been mentioned before. Either expand on why this may be relevant or delete.
Answer Line 263-264: The phrase has been deleted.
Reviewer 2 L267-268 what is this assumption based on? Provide logical thought process and references.
Answer Line 267-268: Investigations of various animal species indicate that the left brain hemisphere is involved in the control of unchanging stimuli or repetitive actions, while the right hemisphere is specialized in the quality of emotional reactions such as aggression or fear, which are accompanied by reduced serotonin secretion. Since serotonin inhibits the impulsive behavior of animals, regulates metabolism, and influences signal transmission between neurons, low blood levels of this neurohormone correlate with the occurrence of depression, aggression, and impulsiveness. Research results suggest that massage may increase vagal nerve activity through stimulation of pressure receptors, which ultimately signal the limbic system and reduce the release of noradrenaline in the bloodstream. Hence, an increase in noradrenaline levels is associated with arousal, stress, and expectation of a reward.
Reviewer 2 L272 there is a body of dog literature that did not find this sex difference in lateral behavior, this should be acknowledged (e.g. Barnard, S., Wells, D. L., Hepper, P. G., & Milligan, A. D. S. (2017, April 17). Association Between Lateral Bias and Personality Traits in the Domestic Dog (Canis familiaris). Journal of Comparative Psychology; Branson, N. J., & Rogers, L. J. (2006). Relationship between paw prefer- ence strength and noise phobia in Canis familiaris. Journal of Compar- ative Psychology, 120, 176–183; Schneider, L. A., Delfabbro, P. H., & Burns, N. R. (2013). Temperament and lateralisation in the domestic dog (Canis familiaris). Journal of Veterinary Behavior: Clinical Applications and Research, 8, 124–134)
Answer: We appreciate the remark made by the Reviewer on the insufficient presentation of results of the research on the absence of a relationship between sex and brain asymmetry in dogs in the Discussion section. Thank you for the search for current literature on this issue. The fragment of the discussion with the literature suggested by the Reviewer has been included in the manuscript. We have added the following sentence: It should be emphasized that there are many studies in which no relationship between the sex of dogs and brain laterality was found (Barnard et al., 2017; Branson et al., 2006; Schneider et al., 2013).
Reviewer 2 L255-257 a comparison with previous studied populations (i.e. did other study fing the same R vs L vs A distribution?) should be briefly discussed here.
Answer: It is believed that Tan (1987) was the first to publish the results of paw preference in dogs. As shown in his research, 57.1% of the studied population was right-preferent, 17.9% were left-preferent, and 25.0% were ambidextrous. The author does not report on differences between the sexes. In turn, Quaranta et al. (2004) have reported that male dogs prefer the left front paw, whereas female dogs (n = 29) show preference for the right paw. In the study conducted by Wells et al., 2018, ambilateral individuals predominated in the tape and Congo tests. As in the present study, there were more individuals with right paw preference than left-pawed animals, but the differences were not statistically significant.
Reviewer 3 Report
Introduction: Need more background information on effect of laterality on stress response, temperament. Line 072: Harmon-Jones research has not been duplicated and results are not somewhat contextual dependent.
Line 077: assumption of right-pawed dogs exhibiting greater adaptation to stress is not supported by the current literature.
Methods: Need more detailed information about procedure of stroking. When exactly in relation to procedure, where (veterinary surgery line 108?) - studies have shown that the carride to the vet clinic is itself stressful for some animals, what location on the dog - head, ears, body, back?, which owner (female, male, primary caregiver, what other confounding variables were present (other family members, exam room, clinic staff?
No behavioral disorders? All of the dogs had zero fear at the vet clinic? Was previous history at clinic controlled for? Age range of 1-13 years is quite large. No previous behavioral training? Does this include obedience training? Professional or by owner?
Icons for male and female in table are off-putting, recommend Male and Female text used instead
Due to previous studies that indicate a sex difference with measured variables, 20 male and 20 female is a small N. Conclusions drawn should be more preliminary with discussion about further research needed. P value is on borderline of significance, again caution drawing conclusions with reported P values AND small sample size.
More acceptable as a preliminary/pilot study for future research
Author Response
COMMENTS TO THE REVIEWER
Thank you for the review; we have responded to the comments in the text. Fragments that have been modified or added are marked with a different color. Irrelevant references have been removed or replaced with appropriate ones. Missing references have been added.
Most of the revisions addressing the review have been made directly next to the Reviewer's comments in the text of the manuscript. They are marked in green.
Reviewer 3
Comments and Suggestions for Authors
Introduction: Need more background information on effect of laterality on stress response, temperament. Line 072: Harmon-Jones research has not been duplicated and results are not somewhat contextual dependent.
Answer Line 072: Thank you very much for pointing out our mistake; the mistake in the literature citations has been corrected in accordance with their original order.
Line 077: assumption of right-pawed dogs exhibiting greater adaptation to stress is not supported by the current literature.
Answer Line 077: The reviewer is right; there is little information to show that right-pawed dogs show greater stress adaptation. Nevertheless, we formulated such a research hypothesis based on the experience and observations reported by other researchers. For instance, the results of studies conducted by Wells et al. (2017) suggest that left-pawed animals show stronger fear responses, are more likely to show aggression, and are less able to cope with stressful situations than right-pawed animals.
A fragment has been added in the text: As reported by Wells et al. (2017), left-pawed animals exhibit stronger fear responses, are more likely to show aggression, and are less able to cope with stressful situations than right-pawed animals.
Wells, D.L.; Hepper, P.G.; Milligan, A.D.S.; Barnard, S. Cognitive bias and paw preference in the domestic dog (Canis familiaris). Journal of Comparative Psychology, 2017, 131, 317-325. https://doi.org/10.1037/com0000080.
Reviewer 3
Methods: Need more detailed information about procedure of stroking. When exactly in relation to procedure, where (veterinary surgery line 108?) - studies have shown that the carride to the vet clinic is itself stressful for some animals, what location on the dog - head, ears, body, back?, which owner (female, male, primary caregiver, what other confounding variables were present (other family members, exam room, clinic staff?
Answer: The dogs were stroked at the second visit immediately before the clinical examination and blood sampling. It was held in a closed 3x3 m room (an unfortunate term “veterinary's surgery” was used in the text) in the absence of other persons. The owner was seated on a chair, and the dog was standing or sitting, whichever he/she preferred. As written in the text, the front part of the neck and the chest were the stroked areas. The stroking duration was determined individually and ranged from 5 to 10 minutes depending on the behavior of the individual animals, i.e. such behavioral symptoms as licking, turning the head away, or restlessness (some dogs needed 2-3 minutes to calm down). The effective stroking and gentle speaking time was 5 minutes.
Only one person was present during the stroking stimulation, i.e. the owner/handler. In our study, the sex, age, and social status of the owner were not regarded as variables influencing the behavior and the level of parameters determined in the serum. The time and method of transporting the animal to the clinic were not taken into account, as this would make it difficult to form an experimental group according to the division into transport by e.g. private car, taxi, or on foot.
These remarks were also provided by another reviewer of this study; therefore, appropriate revisions have been introduced and marked in red in the text.
Reviewer 3
No behavioral disorders? All of the dogs had zero fear at the vet clinic? Was previous history at clinic controlled for? Age range of 1-13 years is quite large. No previous behavioral training? Does this include obedience training? Professional or by owner?
Answer: As for behavioral disorders - the authors wanted to emphasize that previously (in the history of visits) the owner had not reported any symptoms of undesirable behavior or stereotypies. During the appointment at the clinic, there were such behavioral symptoms as licking and turning the head away. They were regarded as typical of the situation. There were no behavioral observations from previous visits, which would have been logistically difficult to carry out on so-called ownership dogs (with an owner/legal guardian) and to conduct the experiment during routine medical-veterinary procedures.
No previous behavioral training - this means that the owner had not used help/services of a professional behaviorist in terms of training and behavioral therapy. The human-animal interaction developed naturally during the everyday life in the household.
Wide age range - in fact, three individuals are responsible for this range: one dog that was 1.2 years old and two older dogs that were 11 years and 12 years old. In other publications on hormonal interactions in dogs, the age of the study group ranged from 1 to 15 years (for example, in Park et al. BMC Veterinary Research 2014, 10, p. 311). The authors did not want to remove the dogs from the experimental group, as the effect of age on the hormonal parameters was not investigated. However, it is reasonable to leave only the average age of 3.91 years and SEM = 0.39 years in the text.
This was also mentioned in the review of another reviewer of this manuscript; therefore, relevant revisions have been made and marked in red in the text.
Reviewer 3
Icons for male and female in table are off-putting, recommend Male and Female text used instead
Answer: Certainly, the symbols were unfortunate and, as recommended by the reviewer, they have been replaced with F for female and M for male in the table.
Reviewer 3
Due to previous studies that indicate a sex difference with measured variables, 20 ma
le and 20 female is a small N. Conclusions drawn should be more preliminary with discussion about further research needed. P value is on borderline of significance, again caution drawing conclusions with reported P values AND small sample size.
Answer: As regards correlations of the assessed parameters with the sex, another reviewer emphasized that “there is a body of dog literature that did not find this sex difference in lateral behavior, this should be acknowledged (e.g. Barnard, S., Wells, D. L., Hepper, P. G., & Milligan, A. D. S. (2017, April 17). Association Between Lateral Bias and Personality Traits in the Domestic Dog (Canis familiaris). Journal of Comparative Psychology; Branson, N. J., & Rogers, L. J. (2006). Relationship between paw prefer- ence strength and noise phobia in Canis familiaris. Journal of Comparative Psychology, 120, 176–183; Schneider, L. A., Delfabbro, P. H., & Burns, N. R. (2013). Temperament and lateralisation in the domestic dog (Canis familiaris). Journal of Veterinary Behavior: Clinical Applications and Research, 8, 124–134)”.
Unfortunately, formation of research groups in the case of ownership dogs (with an owner/legal guardian) and conducting an experiment during routine medical and veterinary activities have certain limitations, e.g. justified hematological re-examination, additional time spent in the clinic on observations, and finally the need for cooperation/active participation of the owner at home (in the dog's place of regular residence) including some so-called sensitive data (sex, age). Therefore, we used the leading literature in the field of brain asymmetry in dogs, which presents research on comparable or smaller sizes of dog groups. For example, 30 individuals were used in the study conducted by Siniscalchi et.al. 2017, 41 individuals were examined by Barnard et al. 2018, and 10 individuals by Handlin et al., 2011. Therefore, we decided to limit the research group to n=40. As for the Reviewer's remark on too bold inference, it can be ascribed to the temperament of the authors.
Marcello Siniscalchi, Serenella d’Ingeo, Serena Fornelli & Angelo Quaranta; Lateralized behavior and cardiac activity of dogs in response to human emotional vocalizations, SCIentIfIC RePOrtS | (2018) 8:77 | DOI:10.1038/s41598-017-18417-4.
Shanis Barnard , Deborah L. Wells and Peter G. Hepper; Laterality as a predictor of coping strategies in dogs entering a rescue shelter, Symmetry 2018, 10, 538; doi:10.3390/sym10110538.
Handlin, L., Hydbring-Sandberg, E., Nilsson, A., Ejdebäck, M., Jansson, A., and Uvnäs-Moberg, K. (2011). Short-term interaction between dogs and their owners – effects on oxytocin, cortisol, insulin and heart rate - an exploratory study. Anthrozoos 24, 301–315. doi: 10.2752/175303711X1304591486 5385.
Reviewer 3
More acceptable as a preliminary/pilot study for future research
Answer: We have tried our best to respond to the reviews. We hope that the revisions introduced in accordance with the Reviewer's suggestion will improve the quality of the manuscript and will qualify our work as worth publishing in Animals.
Reviewer 4 Report
Excellent and creative study design. I was worried at first when I saw a blood draw required but since it was integrated into normal vet care requirements was well done. I have not read much prior research on left vs right paw so this is important research.
I am working on cortisol levels pre and post behavioral training program but using a saliva test So your research was very interesting to me.
Well planned and organized research!
Author Response
Review
Comments and Suggestions for Authors
Excellent and creative study design. I was worried at first when I saw a blood draw required but since it was integrated into normal vet care requirements was well done. I have not read much prior research on left vs right paw so this is important research.
I am working on cortisol levels pre and post behavioral training program but using a saliva test So your research was very interesting to me.
Well planned and organized research!
COMMENTS TO THE REVIEWER
Answer: We would like to thank the Reviewer for the profound study of our research. We are glad that the Reviewer finds our investigations innovative, necessary, well planned and performed, and useful for other researchers studying behavioral laterality. While planning this research, we hoped that it would broaden the current knowledge and allow exchange of experience with other scientists.
Round 2
Reviewer 1 Report
Authors has been taken care to improve the manuscript. It was carried out nearly all recommended revisions. The description of the methods has been supplemented with the necessary information. However, the blood analysis results presented in Tables 2 and 3 are still questionable.
In my first review, I asked the authors about the results of the analysis, but the answer I received did not explain my doubts. Table 2 contain results of level of hormones in the blood serum dogs in relation to the sex and stroking effects (n=40) but Table 3 contain results in relation to laterality and stroking effects (n=40). Hormone levels analysed for sex or laterality are related to the hormones levels before and after stroking. Both tables (rows 8 and 9 in Tab 2 and 3 - Effect of stroking (EB)) should contain the same hormone level values because they contain mean value for the same group of individuals who took part in the experiment (N = 40). In the case of noradrenaline and cortisol, the values are actually the same but the serotonin values are different and to my knowledge this is a mistake.
Mean value of serotonin level under negative and positive stimuli for whole experimental group is ambiguous. According to Table 2 it is 500.12 and 542.75 respectively whereas according to Table 3 it is 530.12 and 592.75 respectively.
Using the mean values from the first four rows of Tables 2 and 3, I calculated the weighted average for effect of sex (EP), effect of stroking (EB), effect of laterality (EL).
There were a few differences between my calculations and the authors' calculations.
In Table 2
Effect of stroking – NS – serotonin – According to my calculation should be 530.12 instead of 500.12
[Mean(M+NS) * count(M+NS) + Mean(F+NS) * count(F+NS)] / (count(M+NS) + count(F+NS)) = (457.73*20 + 607.85 * 24) / (16 + 24) = 530.12 ≠ 500.12
In Table 3:
Weighted average of serotonin level in stroking stimuli should be 575.77 instead of 592.75
[Mean(L + SS) * count(L + SS) + Mean(R + SS) * count(R + SS)] / (count(L + SS) + count(R + SS)) = (527.65 * 16 + 607.85 * 24) / (16 + 24) = 575.77 ≠ 592.75
And respectively:
Weighted average of serotonin level in negative stimuli should be 541.75 instead of 530.12
Effect of laterality – NS – serotonin – Weighted average should be 514.80 instead of 509.80
Effect of laterality – SS – serotonin – Weighted average should be 588.07 instead of 573.01
I would like to receive explanations from the authors regarding the discrepancy in the results. I hope it can be easily explained and possibly corrected.
Author Response
Thank you for the tips in the review. We agree with the comments, and we will try to avoid these shortcomings in the future.
